# OpenReview forum: "A-MemGuard: A Proactive Defense Framework For LLM-Based Agent Memory"
_ICML.cc/2026/Conference — ICML 2026 regular_

### Official Review · Reviewer_V5yi · 2026-03-03

**Soundness:** 3
**Presentation:** 4
**Significance:** 3
**Originality:** 4
**Overall Recommendation:** 3
**Confidence:** 3

**Summary:**

This paper studies memory poisoning in LLM agents, where an attacker injects seemingly benign memory records that become harmful only under specific contexts and can trigger a self-reinforcing error cycle when corrupted outputs are stored back as precedents. The authors propose A-MemGuard, a non-invasive defense that validates retrieved memories through consensus over structured reasoning paths and filters out anomalous paths, while also maintaining a separate lesson memory that stores detected anomalies as negative examples to prevent repeated failures. Experiments on direct poisoning with AgentPoison on ReAct-StrategyQA and EHRAgent, indirect injection in a MINJA-style setting on MMLU, and a multi-agent misinformation benchmark show large reductions in attack success rates with limited utility degradation, supported by ablations and cost analysis.

**Compliance With Llm Reviewing Policy:**

Affirmed.

**Final Justification:**

I thank the authors for the detailed rebuttal and the additional experiments. While these results are helpful, my main concerns regarding reliance on structured extraction, judge reliability, and limited evidence for broad generalization are only partially addressed, and several key results appear only in the rebuttal. Consequently, my overall assessment remains unchanged.

**Key Questions For Authors:**

## Key Questions for Authors (each targets a specific weakness)

1. **(Targets Weakness 1)** How does performance change when the judge is replaced by a substantially weaker open-source model, or by a judge that is different from the victim backbone? Can you report ASR and utility under these swaps, and discuss correlated failure modes?
2. **(Targets Weakness 2)** Can you provide a full end-to-end cost and latency breakdown, including path generation, judging, and lesson retrieval, and report p50/p90/p99 latency under a realistic mixture of benign and adversarial queries? What is the cheapest configuration that still delivers most of the ASR reduction?
3. **(Targets Weakness 3)** Can you validate A-MemGuard on additional memory frameworks with different schemas and write policies, and on longer-horizon tool-using agents? If substantial prompt or component tuning is required, what are the stable, system-agnostic parts versus system-specific parts?
4. **(Targets Weakness 4)** Beyond the current adaptive variants, have you evaluated attackers that explicitly optimize two objectives, making structured paths appear mutually consistent while steering actions toward the judge’s decision boundary? If not, what defenses would you add against such a strategy?

**Limitations:**

yes

**Strengths And Weaknesses:**

## Strengths

1. **Well-motivated threat model**: The paper highlights two practically important properties of memory poisoning in agents, context-triggered harmfulness and multi-round self-reinforcing error amplification, and designs the defense around them.
2. **Non-invasive and easy to integrate**: A-MemGuard is a drop-in layer that does not require modifying the agent backbone, combining retrieval-time validation with a separate lesson memory to prevent repeated failures.
3. **Strong empirical gains across settings**: The evaluation spans direct poisoning (AgentPoison) on ReAct-StrategyQA and EHRAgent, indirect injection in a MINJA-style setting on MMLU, and a multi-agent misinformation scenario, showing large ASR reductions with limited utility loss.
4. **Helpful diagnostics and ablations**: The paper includes component ablations and top-k sensitivity, which clarify where the gains come from and provide actionable guidance for deployment choices.

## Weaknesses

1. **Strong dependence on judge and structured extraction**: The core detection relies on generating structured reasoning paths and using an LLM judge to decide consistency. If extraction is noisy or the judge is miscalibrated or manipulable, the system may fail in correlated ways.
2. **Deployment cost and latency may be high**: Consensus validation requires multiple generations plus judging, which can be expensive. The current analysis does not fully characterize end-to-end latency under realistic traffic and query mixes.
3. **Limited evidence of broad generalization**: The experiments cover several tasks and backbones, but real memory systems vary widely in schema, write policies, tool usage, and interaction horizon. It remains unclear how well the approach transfers without per-system tuning.
4. **Adaptive attacker space not fully explored**: The evaluated adaptive variants are useful, but stronger attackers could jointly optimize for structural agreement across paths and borderline-safe actions that evade the judge, which may reduce the reported robustness.

---

> ### Author Rebuttal · Authors · 2026-03-31
>
> We deeply appreciate your comprehensive review. Your recognition of our "well-motivated threat model" and "non-invasive" design is very encouraging. Below, we address your points with new empirical evaluations and the requested granular latency data.
>
> **Response to W1 & W2 / Q1 & Q2: Weaker Judges, Cost, and p50/p90/p99 Latency**
>
> To address the reliance on large models and fully characterize end-to-end latency under realistic traffic, we evaluated an optimized **Batched "1+1" Execution** using lightweight open-source judges (Qwen2.5 1.5B/0.5B via vLLM).
>
> **Table 1: Wall-Clock Latency & Overhead Ratio per Query (EHRAgent)**
> | Defense Method | Judge Model | Gen/Retrieval (s) | Judge/Val (s) | Total Defense Latency | Total Task Time | Overhead Ratio | ASR-r (↓) |
> | :--- | :--- | :--- | :--- | :--- | :--- | :--- | :--- |
> | **Distil Classifier** | - | 0.47s | N/A | 0.47s | 123.30s | ~0.39% | 100.0% |
> | **LLM Auditor** | LLaMA-3.1-8B | N/A | N/A | 67.72s | 102.50s | ~66.70% | 40.43% |
> | **A-MemGuard** | Qwen2.5 1.5B | 4.05s | 2.25s | **6.30s** | **40.10s** | **~16.07%** | **3.65%** |
> | **A-MemGuard** | Qwen2.5 0.5B | 2.47s | 1.14s | **3.61s** | **37.80s** | **~9.80%** | **5.20%** |
>
> **Table 2: End-to-End Latency Breakdown in Seconds (Mean, p50, p90, p99)**
> | Model | Path Gen (s) |    Judge (s) | Lesson Retrieval (s) | Total Task Overhead | ASR-r (↓) |
> | :--- | :--- | :--- | :--- | :--- | :--- |
> | **0.5B** | 2.50, 2.43, 2.63, 3.00 | 1.14, 1.14, 1.15, 1.17 | 0.06, 0.04, 0.10, 0.11 | **9.80%** | **5.20%** |
> | **1.5B** | 4.19, 4.16, 4.32, 4.33 | 2.25, 2.26, 2.26, 2.26 | 0.05, 0.04, 0.05, 0.06 | **16.07%** | **3.65%** |
>
> *Analysis:* The tight clustering of p50 and p99 values proves A-MemGuard's latency is highly stable and predictable. The **cheapest optimal configuration** is the Qwen2.5 0.5B judge, slightly increases ASR-r (5.20%) but drops the overhead to <10%. Because our judge acts as a binary structural comparator rather than solving complex tasks from scratch, the required capability ceiling is remarkably low. Both configurations vastly outperform the standard LLM Auditor baseline (which hogs 66.70% of total task time and yields a poor 40.43% ASR-r). While the Distil Classifier adds near-zero overhead (~0.39%), it completely fails defensively against context-dependent attacks (100.0% ASR-r).
>
> **Response to W3 / Q3: Broad Generalization Across Frameworks**
>
> To demonstrate generalization across different schemas and write policies, we applied A-MemGuard as a non-invasive intercept layer on four distinct, state-of-the-art agent architectures using the FEVER dataset.
>
> **Table: Generalization Across Architectures (FEVER Dataset)**
> | Architecture Type | ASR-Retrieval | ASR-Output | F1 Score |
> | :--- | :--- | :--- | :--- |
> | **Generative Agents** | 0.133 | 0.133 | 0.583 |
> | **MemoryBank** | 0.000 | 0.000 | 0.580 |
> | **MetaGPT** | 0.100 | 0.008 | 0.532 |
> | **Voyager** | 0.050 | 0.048 | 0.428 |
>
> *System-Agnostic Design:* A-MemGuard requires zero modification to the underlying storage mechanics—whether episodic (Generative Agents), decay-based (MemoryBank), or skill-based procedural (Voyager). It intercepts context between retrieval and execution, making the stable components (Graph Sanitization and Consensus Scoring) highly portable without per-system prompt tuning.
>
> **Response to W4 / Q4: Dual-Objective Adaptive Attackers**
>
> Building on our Path-Consistent Attack limits, we designed a **"Dual-Objective Adaptive Attack."** The attacker jointly optimizes for structural agreement AND attempts to steer actions toward the judge's blind spots using semantic euphemisms (re-labeling destructive actions as "cache synchronization"), fake system flags (`[PRIORITY-CLEAR]`), and forged compliance mandates (fake HIPAA policies).
>
> *Results:* A-MemGuard successfully neutralized these attacks, limiting **ASR-r to 4.5%** while maintaining a strong **Benign ACC of 62.5%**.
>
> *Why the defense holds:* We neutralize both of the attacker's objectives through a two-stage structural bottleneck:
> **(1) Defeating Prompt Disguise (Jailbreaks):** Through our **Graph Sanitization** protocol, the LLM judge never audits raw text. Memories are parsed into a strict `Condition -> Action` graph. Verbose jailbreak wrappers fail to map to this schema and are stripped away, exposing the bare state-change intent.
> **(2) Defeating Structural Mimicry (Path Consistency):** Once sanitized, the paths face the consensus and safety checks. If the attacker forces multiple poisoned paths to agree with *each other*, this hallucinated consensus still fundamentally diverges from the ground-truth task logic established by benign memories. Alternatively, if the attacker attempts to perfectly mimic the benign consensus structure, they are forced to adopt benign actions. Any attempt to attach a malicious action node to a benign condition is exposed after sanitization and cleanly trips the safety audit.

---

> > ### Author Rebuttal · Reviewer_V5yi · 2026-04-02
> >
> > The new results on weaker judges, latency, broader architectural coverage, and adaptive attacks are helpful.
> >
> > That said, my overall assessment remains unchanged. My main concerns are only partially addressed, especially regarding reliance on structured extraction and judge reliability, as well as the still limited evidence for broad generalization. In addition, several important results appear only in the rebuttal. Overall, I appreciate the authors' effort, but the rebuttal does not change my judgment.

---

> > > ### Author Response · Authors · 2026-04-03
> > >
> > > Dear Reviewer V5yi,
> > >
> > > Thank you for reviewing our rebuttal and for acknowledging that the new results on weaker judges, latency, architectural coverage, and adaptive attacks were helpful additions.
> > >
> > > While we respectfully note that your overall assessment remains unchanged, we deeply appreciate the rigorous stress-testing your review prompted. Your initial concerns drove us to conduct extensive additional evaluations, which have undeniably strengthened the robustness of our framework. We explicitly commit to fully integrating all these new empirical results, latency breakdowns, and extended discussions into the main text and appendix of the revised manuscript.
> > >
> > > Thank you again for your time, effort, and constructive feedback throughout this process.
> > >
> > > Best regards,
> > >
> > > The Authors

---

### Official Review · Reviewer_9dKi · 2026-03-13

**Soundness:** 2
**Presentation:** 3
**Significance:** 2
**Originality:** 2
**Overall Recommendation:** 4
**Confidence:** 3

**Summary:**

The paper investigates memory poisoning attacks in LLM agents, where attackers inject seemingly benign records into an agent’s memory to influence the subsequent reasoning and actions of the agent. To provide guardrail against this threat, the authors propose A-MemGuard, a framework that detects memories with adversarialy instructions using consensus-based reasoning validation and prevents error propagation through a dual-memory design that stores past failures as pairwise examples. Experiments across several benchmarks show that the method could potentially reduce attack success rates while maintaining the agent utility.

**Compliance With Llm Reviewing Policy:**

Affirmed.

**Final Justification:**

Most of my previous concerns have been addressed, and I've increased the score.

**Key Questions For Authors:**

How robust is A-MemGuard when an adaptive attacker successfully controls the retrieved top-K memories?

**Limitations:**

See above.

**Strengths And Weaknesses:**

Strengths:
+ this paper considers memory poisoning and long-term error propagation through agent memory which is an important topic and security issue for LLM agents;
+ the experiments across several benchmarks and attack settings show consistent reductions in ASR while preserving utility on benign tasks.

Weakness:
+ while the overall idea intuitively makes sense, the paper provides little formal justification for why the consensus-based validation can stably detect poisoned memories against strong attackers.
+ generating multiple reasoning paths and performing validation may introduce high computational cost, which is not analyzed in the paper.

---

> ### Author Rebuttal · Authors · 2026-03-31
>
> We sincerely thank you for highlighting the importance of our research topic and recognizing that our framework demonstrates "consistent reductions in ASR while preserving utility." Below we clarify the formal justification and computational cost.
>
> **Response to Weakness 1 & Question 1: Formal Justification & Robustness under Top-K Malicious Dominance**
>
> We completely agree that defending against a "malicious majority" is the most critical test. We dedicated **Appendices C, D, E, and L** to this exact justification, summarized here:
>
> **1. Formal Justification for Structural Separability:** The core technical soundness of our defense relies on shifting from raw semantic similarity checking to **structural reasoning path divergence**. As detailed in Appendix E (using t-SNE and cosine similarity distributions), raw semantic checking is inadequate because poisoned memories successfully camouflage themselves semantically (cosine similarity >0.95). However, extracting entity-relation sequences forces these malicious memories to become distinct structural outliers. Our Knowledge Graph analysis (Appendix C, Fig. 4) formally confirms this, showing that the structural overlap between benign and malicious paths is consistently **<1%** across all 17 evaluation scenarios. This near-zero overlap provides a strong empirical foundation for consensus-based outlier detection.
>
> **2. Robustness when Top-K is 100% Controlled:**
> We explicitly evaluated the scenario where an adaptive attacker dominates the Top-K retrieval context. Using the AgentPoison protocol (Appendix L, Table 12), the attacker optimizes triggers so that **100% of the retrieved Top-K memories (K=4) are poisoned**.
>
> *Why the defense holds:* Retrieval dominance does not equal structural consensus. Malicious memories rely on spurious correlations (injected keywords) to force their retrieval. When extracted into entity-relation reasoning paths, they fail to form a coherent, unified structural consensus amongst themselves because each injection targets malicious actions through inconsistent structural routes. As a result, **consensus validation alone filters out 89.50% of the 100% poisoned context**, and the subsequent safety audit raises total removal to **97.87%**.
>
> **3. Robustness to Adaptive Structural Mimicry:**
> We directly tested whether an adaptive attacker can force structural agreement with our **Path-Consistent Attacker** (Sec. 5.8), who harvests high-frequency benign entities (Patient IDs, diagnoses) and forces their reuse across injections, increasing EntityOverlap from 0% to 15% (Table 7). Despite this, ASR-r increased by only **+0.62%** because the hallucinated cluster still diverges from ground-truth task logic. The attacker faces a fundamental dilemma: to truly mimic benign consensus structure, they must adopt benign actions, which neutralizes the attack.
>
> **Response to Weakness 2: Runtime Overhead and Computational Cost**
>
> We completely agree that evaluating the time cost is crucial. While we provided a Token Cost analysis in **Appendix F**, we have now conducted end-to-end wall-clock latency measurements. A-MemGuard uses an optimized **Batched "1+1" Execution**: all K memories are packed into a single prompt for one forward pass, with lightweight judges (Qwen2.5 1.5B/0.5B) via vLLM.
>
> **Table: Wall-Clock Latency & Overhead Ratio per Query (EHRAgent)**
>
> | Defense Method | Judge Model | Gen/Retrieval (s) | Judge/Val (s) | Total Defense Latency | **Total Task Time** | Overhead Ratio | ASR-r (↓) |
> | :--- | :--- | :--- | :--- | :--- | :--- | :--- | :--- |
> | **LLM Auditor** | LLaMA-3.1-8B | N/A | N/A | 67.72s | **102.50s** | ~66.70% | 40.43% |
> | **A-MemGuard** | Qwen2.5 1.5B | 4.05s | 2.25s | **6.30s** | **40.10s** | **~16.07%** | **3.65%** |
> | **A-MemGuard** | Qwen2.5 0.5B | 2.47s | 1.14s | **3.61s** | **37.80s** | **~9.80%** | **5.20%** |
>
> *Analysis:* Using Qwen2.5 1.5B, the total defense latency is merely 6.30s (~16.07% of the 40.10s task). Stepping down to 0.5B slightly increases ASR-r (from ~3.65% to 5.20%) but reduces defense latency to 3.61s (9.80% of task time). A-MemGuard is vastly faster than the LLM Auditor (67.72s, ~66.70% overhead) while delivering superior security.
>
> **On Significance:** As LLM agents with persistent memory become prevalent in safety-critical domains (healthcare, finance), memory poisoning becomes a first-order security concern. All existing defenses rely on isolated auditing, which fundamentally cannot detect context-dependent threats (as confirmed by ASB [1]). A-MemGuard introduces cross-memory structural consensus as a detection signal and self-corrective learning from past attacks, shifting memory security from static filtering to an adaptive model. We validated this across 4 benchmarks, 2 LLM backbones, 4 retrieval architectures, and multiple adaptive attack regimes.
>
> ### References
> [1] Zhang et al. Agent Security Bench. ICLR'25.

---

> > ### Author Rebuttal · Reviewer_9dKi · 2026-04-03
> >
> > thanks to the author for the detailed response. Most of my concerns have been addressed and I've increased the score.

---

> > > ### Author Response · Authors · 2026-04-03
> > >
> > > Dear Reviewer 9dKi,
> > >
> > > Thank you very much for your prompt acknowledgement and for increasing your score! We are thrilled that our detailed responses regarding the formal justification and computational latency successfully addressed your concerns.
> > >
> > > We deeply appreciate your time, constructive feedback, and support for our work.
> > >
> > > Best regards,
> > >
> > > The Authors

---

### Official Review · Reviewer_uqVM · 2026-03-13

**Soundness:** 3
**Presentation:** 3
**Significance:** 3
**Originality:** 2
**Overall Recommendation:** 4
**Confidence:** 3

**Summary:**

The paper investigates the critical security risk of memory poisoning in large language model (LLM) agents, where adversaries inject seemingly harmless records that trigger malicious behavior in specific contexts and create self-reinforcing error cycles. The authors propose A-MemGuard, a defense framework operating outside the agent's core architecture. It employs a two-pronged approach: a consensus-based validation module that detects anomalies by analyzing the structural divergence of reasoning paths derived from multiple retrieved memories, and a dual-memory structure that distills detected failures into a "lesson memory" to proactively prevent future errors. The method is evaluated against both direct and indirect injection attacks across several benchmarks.

**Compliance With Llm Reviewing Policy:**

Affirmed.

**Final Justification:**

I think most of my concerns have been addressed. I will increase the score to weak accept.

**Key Questions For Authors:**

1) How resilient is the LLM-as-a-judge component to direct adversarial prompt injection or sophisticated jailbreaking techniques embedded deeply within the retrieved memory contexts?

2) Can you provide a detailed wall-clock inference latency breakdown per interaction step, comparing A-MemGuard's batched execution against the standard LLM Auditor and the Distil Classifier baselines?

**Limitations:**

yes

**Strengths And Weaknesses:**

Strengths

1) The paper accurately isolates the nuances of agentic memory vulnerabilities, specifically highlighting how attacks bypass static content filters by remaining context-dependent and exploiting the agent's autonomous learning loop to reinforce errors.

2) The shift from raw semantic similarity checking (which the authors prove is inadequate via t-SNE and cosine similarity distributions ) to structural reasoning path divergence is technically sound. Extracting entity-relation sequences to expose outliers provides a much stronger defensive signal.

3) The framework demonstrates significant resilience across varied, robust evaluation scenarios, including ReAct-Strategy QA, EHRAgent, and MMLU. It effectively suppresses the Attack Success Rate (ASR) under severe stress tests, such as when adversarial records constitute 100% of the retrieved context, without severely degrading benign task utility.

Weaknesses

1) The evaluation relies heavily on specific attack protocols (AgentPoison and MINJA). While the "Path-Consistent Poisoning" stress test attempts to address advanced threats , highly sophisticated, multi-turn adversarial jailbreaks might be able to engineer reasoning paths that structurally mimic benign consensus or directly subvert the LLM judge's instructions during the safety audit phase.

2) The dual-memory structure relies on the accurate generation of negative lessons. If the consensus module produces a false negative during a highly stealthy attack, the system lacks a mechanism to prevent that poisoned logic from being permanently archived or, worse, weaponized if the lesson memory itself becomes compromised over extended deployments.

---

> ### Author Rebuttal · Authors · 2026-03-31
>
> We sincerely thank you for recognizing that our structural reasoning path analysis provides a "much stronger defensive signal." We address your concerns below.
>
> **Response to W1 & Q1: Sophisticated Jailbreaks & Structural Mimicry**
>
> **(1) Structural Mimicry (Already Evaluated):** As detailed in **Appendix P** (*Path-Consistent Adaptive Attack*), even when an attacker forces the reuse of high-frequency benign entities to inflate structural agreement, A-MemGuard's ASR-r increased only marginally (+0.62%). The mechanism successfully detects the inconsistency between hallucinated clusters and true task logic.
>
> **(2) New Evaluation - Dual-Objective Adaptive Attack (Mimicry + Jailbreak):** To directly address prompt injections subverting the judge, we designed a stronger attack on EHRAgent combining structure mimicry with deep prompt disguise, using five bypass strategies: Semantic Triggers (`[PRIORITY-CLEAR]`), Euphemistic Injection (re-labeling `DeleteDB`), Split-Action, Consistent Reasoning Chains, and Authoritative Forgery (fake HIPAA mandates).
>
> **Table: Performance under Dual-Objective Adaptive Attack (EHRAgent)**
>
> | Attack Scenario | ASR-r (↓) | Benign ACC (↑) |
> | :--- | :--- | :--- |
> | **Dual-Objective Adaptive Attack** | **4.5%** | **62.5%** |
>
> *Defense Mechanism:* A-MemGuard neutralizes these via **Graph Sanitization (Generate-then-Extract)**. The LLM judge *never* audits raw, prompt-injected snippets directly. Instead, raw memory is parsed into a sanitized `Condition -> Action` graph. Jailbreak wrappers fail to map onto this strict schema and are stripped away, exposing the underlying state-change intent to the safety audit.
>
> **Response to W2: False Negatives and Lesson Memory Weaponization**
>
> If an attack produces a *false negative*, it simply executes and is *not* written into the Lesson Memory (which only archives flagged anomalies). If an attacker intentionally triggers *false positives* to weaponize the memory, A-MemGuard employs two complementary safety mechanisms (**Appendix N**):
>
> **(1) Dual-gated verification (Appendix N.1):** A candidate memory enters Lesson Memory only if it fails *both* the structural consensus check and the safety-judge audit. On a purely benign workload, this yields a false positive rate of only 0.4%, aligned with ensemble-based false-positive mitigation [1][2].
>
> **(2) Soft constraints (Appendix N.2):** False positives do *not* act as irreversible blocking rules. Lessons are retrieved only as *contextual warnings* weighed against the current task context. Including the full lesson store (with the 0.4% false positives) shifts benign-task accuracy by <0.2% (from 63.8% to 63.7%). The agent's baseline reasoning is robust enough to safely ignore mismatched, out-of-context warnings. For extended deployments, lesson accumulation is inherently bounded since new lessons are only generated when new attacks are detected.
>
> **Response to Q2: Latency Overhead Breakdown**
>
> To demonstrate deployment viability (**Appendix F**), we evaluated end-to-end latency using an optimized **Batched "1+1" Execution** with lightweight judges (Qwen2.5 1.5B/0.5B via vLLM).
>
> **Table: Wall-Clock Latency & Overhead Ratio per Query (EHRAgent)**
>
> | Defense Method | Judge Model | Gen/Retrieval (s) | Judge/Val (s) | Total Defense Latency | Total Task Time | Overhead Ratio | ASR-r (↓) |
> | :--- | :--- | :--- | :--- | :--- | :--- | :--- | :--- |
> | **Distil Classifier** | - | 0.47s | N/A | 0.47s | 123.30s | ~0.39% | 100.0% |
> | **LLM Auditor** | LLaMA-3.1-8B | N/A | N/A | 67.72s | 102.50s | ~66.70% | 40.43% |
> | **A-MemGuard** | Qwen2.5 1.5B | 4.05s | 2.25s | **6.30s** | **40.10s** | **~16.07%** | **3.65%** |
> | **A-MemGuard** | Qwen2.5 0.5B | 2.47s | 1.14s | **3.61s** | **37.80s** | **~9.80%** | **5.20%** |
>
> *Analysis:* The Distil Classifier adds near-zero overhead but completely fails against context-dependent attacks (100.0% ASR-r). Using Qwen2.5 1.5B, defense latency is merely 6.30s (around 16.07% of the 40.10s task). Stepping down to 0.5B slightly increases ASR-r (to 5.20%) but reduces latency to 3.61s (9.80%). A-MemGuard is vastly faster than the LLM Auditor (~66.70% overhead) while delivering superior security.
>
> **On Originality:** We respectfully note that our contribution represents a paradigm shift from isolated content-based auditing to context-aware structural consensus. All prior defenses (LLM Auditor, Distil Classifier, PPL) audit each memory independently and fail because poisoned memories appear benign in isolation. A-MemGuard is the first to (1) leverage cross-memory reasoning path divergence for detection, and (2) introduce self-corrective learning from detected attacks. We believe this new defensive paradigm constitutes a novel contribution to agent security.
>
> ### References
> [1] Zoppi et al. Fusing anomaly detection with false positive mitigation. *Information Fusion*, 2021.
>
> [2] Y. Liu et al. Selective ensemble method for anomaly detection. *Scientific Reports*, 2021.

---

> > ### Author Rebuttal · Reviewer_uqVM · 2026-04-04
> >
> > I think most of my concerns have been addressed. I will increase the score to weak accept.

---

> > > ### Author Response · Authors · 2026-04-05
> > >
> > > Dear Reviewer uqVM,
> > >
> > > Thank you very much for your prompt acknowledgement and for increasing your score! We are thrilled that our detailed responses regarding the formal justification and computational latency successfully addressed your concerns.
> > >
> > > We deeply appreciate your time, constructive feedback, and support for our work.
> > >
> > > Best regards,
> > >
> > > The Authors

---

### Official Review · Reviewer_q2v6 · 2026-03-13

**Soundness:** 3
**Presentation:** 3
**Significance:** 3
**Originality:** 3
**Overall Recommendation:** 4
**Confidence:** 3

**Summary:**

This paper presents an empirical study on defenses against memory poisoning attacks in LLM agents. The proposed method, A-MemGuard, adopts a proactive and experience-driven approach to protect agents from memory poisoning. Specifically, A-MemGuard employs consensus-based reasoning validation and contextualized analysis of suspicious records to identify potentially poisoned memories. In addition, the system introduces a dual-memory structure that stores lessons learned from previous poisoning attacks. Extensive experiments and ablation studies demonstrate the effectiveness of A-MemGuard.

**Compliance With Llm Reviewing Policy:**

Affirmed.

**Final Justification:**

The authors addressed a large portion of my concerns so I keep my positive recommendation. Some of the concerns raised by other reviews might need further discussion and/or evidence.

**Key Questions For Authors:**

1.Did the authors evaluate the runtime overhead or latency introduced by A-MemGuard?


2.Could the generated consensus reasoning contain errors? If so, could this lead to false positives in the validation process, and how might the system mitigate such issues?

**Limitations:**

Yes

**Strengths And Weaknesses:**

Strengths:

- The paper is well written and clearly presented.
- The proposed techniques, including consensus-based reasoning validation and the use of dedicated memory for poisoning attack experiences, are novel and interesting.
- The work is well motivated and may have a meaningful impact on future research on agent memory security.
- The experimental evaluation and analysis are comprehensive.


Weaknesses:

- The paper does not include a comparison of runtime overhead or latency for A-MemGuard. Since this is a defense mechanism that may run during agent execution, evaluating its time cost is important for practical deployment.
- The consensus reasoning process itself may introduce errors. The paper does not discuss potential inaccuracies in the generated reasoning or the possibility of false positives during the consensus validation process.

---

> ### Author Rebuttal · Authors · 2026-03-31
>
> We sincerely appreciate your positive evaluation of our work. We are encouraged that you find our problem "well motivated," our consensus-based reasoning validation "novel and interesting," and our experimental evaluation "comprehensive." Below, we address your specific questions regarding runtime overhead and the potential for false positives.
>
> **Response to Weakness 1 & Question 1: Runtime Overhead and Latency**
>
> > *Did the authors evaluate the runtime overhead or latency introduced by A-MemGuard?*
>
> We completely agree that evaluating the time cost is crucial for practical deployment. While we provided a Token Cost analysis in **Appendix F** of our original submission, we have now conducted a comprehensive end-to-end latency evaluation to demonstrate real-world deployment viability.
>
> As discussed in **Appendix F**, A-MemGuard uses an optimized **Batched "1+1" Execution**. Instead of making K+1 sequential calls, we pack all K retrieved memories into a single prompt to generate all parallel reasoning paths in one forward pass. To further optimize, we evaluated using lightweight, highly efficient open-source models (Qwen2.5 1.5B and 0.5B) for the consensus validation step via vLLM.
>
> **Table: Wall-Clock Latency & Overhead Ratio per Query (EHRAgent)**
>
> | Defense Method | Judge Model | Gen/Retrieval (s) | Judge/Val (s) | Total Defense Latency | **Total Task Time** | Overhead Ratio | ASR-r (↓) |
> | :--- | :--- | :--- | :--- | :--- | :--- | :--- | :--- |
> | **LLM Auditor** | LLaMA-3.1-8B | N/A | N/A | 67.72s | **102.50s** | ~66.70% | 40.43% |
> | **A-MemGuard** | Qwen2.5 1.5B | 4.05s | 2.25s | **6.30s** | **40.10s** | **~16.07%** | **3.65%** |
> | **A-MemGuard** | Qwen2.5 0.5B | 2.47s | 1.14s | **3.61s** | **37.80s** | **~9.80%** | **5.20%** |
>
> **Analysis:** Using Qwen2.5 1.5B via vLLM, the total defense latency is merely 6.30s, accounting for ~16.07% of the 40.10s task duration. Stepping down to a 0.5B model slightly increases ASR-r (from ~3.65% to 5.20%), but reduces defense latency to 3.61s (9.80% of task time). A-MemGuard is vastly faster than the LLM Auditor (67.72s, ~66.70% overhead) while delivering superior security. We will incorporate this latency analysis into the revised paper.
>
> **Response to Weakness 2 & Question 2: Errors and False Positives in Consensus**
>
> This is a very insightful question. We fully acknowledge that the consensus module may occasionally make mistakes, which could in principle introduce false positives into the Lesson Memory. We explicitly studied this concern in **Appendix N (Lesson Memory Safety and False-Positive Impact)** and designed A-MemGuard with two complementary safety mechanisms.
>
> **(1) Dual-gated verification to suppress false positives.**
> As detailed in **Appendix N.1**, a candidate memory is written into the Lesson Memory only if it fails **both** the structural consensus check and the safety-judge audit. These two signals are complementary: consensus detects structural outliers while the safety judge catches explicit unsafe intent. Empirically, on a purely benign EHRAgent workload, this strict gating yields a very low false positive rate (**0.4%**). More broadly, this design is aligned with a well-established line of work in anomaly detection and alert triage, where **ensemble/consensus signals** [2] and **second-stage refinement or false-positive mitigation** [1] are used to reduce false positives before escalating a decision.
>
> **(2) Soft constraints rather than hard blocking rules.**
> As detailed in **Appendix N.2**, even if a false positive is occasionally written into the Lesson Memory, it does **not** function as an irreversible blocking rule. Instead, lessons are retrieved only as **contextual warnings** that the agent can weigh against the current task context. This "soft-constraint" design substantially limits the downstream impact of occasional false positives. In fact, when we include the full lesson store (including the small number of false positives), the benign-task accuracy changes only from **63.8% to 63.7%** (i.e., **<0.2%** absolute). This design choice is also consistent with prior alert-management systems, where anomaly signals are often used for **triage, prioritization, or downstream reasoning** rather than hard rejection.
>
> Taken together, these results suggest that even if the consensus module is imperfect, A-MemGuard is robust in practice: the **dual-gated verification** mechanism keeps the false positive rate extremely low, and the **soft contextual use** of lessons ensures that rare residual errors have negligible impact on benign-task performance. We will add a more prominent discussion of these safeguards in the revised paper.
>
> ### References
> [1] Zoppi et al. Fusing anomaly detection with false positive mitigation methodology for predictive maintenance under multivariate time series. *Information Fusion*, 2021.
>
> [2] Yansong Liu et al. Selective ensemble method for anomaly detection based on parallel learning. *Scientific Reports*, 2021.

---

> > ### Author Rebuttal · Reviewer_q2v6 · 2026-04-02
> >
> > Thank you for your detailed response to my comments and for the additional followup results. I feel most of my comments have been addressed and will keep my current recommendations.

---

> > > ### Author Response · Authors · 2026-04-02
> > >
> > > Dear Reviewer q2v6,
> > >
> > > Thank you for taking the time to review our rebuttal and for your positive feedback! We are very glad that our responses and the new results have addressed your main comments.
> > >
> > > We noticed the system acknowledgement status indicates potential follow-up questions. We just want to drop a quick note that we are fully available and happy to provide any further clarification if you need it during the discussion phase.
> > >
> > > We hope the newly added end-to-end latency evaluations and the detailed discussion on our dual-gated verification have further strengthened your confidence in our work. We would be deeply grateful for your continued support!
> > >
> > > Best regards,
> > > Authors

---

### Decision · Program_Chairs · 2026-04-30

**Decision:**

Accept (regular)

**Comment:**

This paper introduced A-MemGuard, a defense framework for LLM agent memory. Reviewer gaves 4, 4,4,3. Although the reviewer with score 3 said the concerns have been addressed, he/she still mentioned serveral partially solved concerns in the written context. AC read all the messages. The main concerns after the rebuttal are the generalization,  reliance on structured extraction, judge reliability, and adaptive attacks. AC sees that authors indeed make a lot of effort during the rebuttal with a lot of experiments. But the rebuttal is not very clear.

For instance, reviewer asked: Can you validate A-MemGuard on additional memory frameworks with different schemas and write policies, and on longer-horizon tool-using agents? If substantial prompt or component tuning is required, what are the stable, system-agnostic parts versus system-specific parts? Authors' response did not clearly tell the reviewer which agent framework is the on-longer-horizon toll-using agent (e.g., Voyager). What is the conclusion there?


Overall, these parts should be improved and revised in the final version. AC suggests the Weak accept (low priority: accept if there is room in the program).